# An Integrative Computational Approach for Identifying Cotton Host Plant MicroRNAs with Potential to Abate CLCuKoV-Bur Infection

**DOI:** 10.3390/v17030399

**Published:** 2025-03-12

**Authors:** Muhammad Aleem Ashraf, Imran Shahid, Judith K. Brown, Naitong Yu

**Affiliations:** 1Institute of Tropical Biosciences and Biotechnology, Chinese Academy of Tropical Agricultural Sciences, Haikou 571101, China; 2Department of Biosciences and Technology, Emerson University, Multan 60000, Pakistan; 3Department of Pharmacology and Toxicology, Faculty of Medicine, Umm Al-Qura University, Makkah 21955, Saudi Arabia; iyshahid@uqu.edu.sa; 4School of Plant Sciences, The University of Arizona, Tucson, AZ 85721, USA; jbrown@ag.arizona.edu

**Keywords:** *begomovirus*, binding affinity, cotton leaf curl disease resistance, gene silencing, in silico tools, RNAi, R-language, target prediction

## Abstract

*Cotton leaf curl Kokhran virus*-Burewala (CLCuKoV-Bur) has a circular single-stranded ssDNA genome of 2759 nucleotides in length and belongs to the genus *Begomovirus* (family, *Geminiviridae*). CLCuKoV-Bur causes cotton leaf curl disease (CLCuD) and is transmitted by the whitefly *Bemisis tabaci* cryptic species. Monopartite begomoviruses encode five open reading frames (ORFs). CLCuKoV-Bur replicates through a dsDNA intermediate. Five open reading frames (ORFs) are organized in the small circular, single-stranded (ss)-DNA genome of CLCuKoV-Bur (2759 bases). RNA interference (RNAi) is a naturally occurring process that has revolutionized the targeting of gene regulation in eukaryotic organisms to combat virus infection. The aim of this study was to elucidate the potential binding attractions of cotton-genome-encoded microRNAs (*Gossypium hirsutum*-microRNAs, ghr-miRNAs) on CLCuKoV-Bur ssDNA-encoded mRNAs using online bioinformatics target prediction tools, RNA22, psRNATarget, RNAhybrid, and TAPIR. Using this suite of robust algorithms, the predicted repertoire of the cotton microRNA-binding landscape was determined for a CLCuKoV-Bur consensus genome sequence. Previously experimentally validated cotton (*Gossypium hirsutum* L.) miRNAs (*n* = 80) were selected from a public repository miRNA registry miRBase (v22) and hybridized in silico into the CLCuKoV-Bur genome (AM421522) coding and non-coding sequences. Of the 80 ghr-miRNAs interrogated, 18 ghr-miRNAs were identified by two to four algorithms evaluated. Among them, the ghr-miR399d (accession no. MIMAT0014350), located at coordinate 1747 in the CLCuKoV-Bur genome, was predicted by a consensus or “union” of all four algorithms and represents an optimal target for designing an artificial microRNA (amiRNA) silencing construct for in planta expression. Based on all robust predictions, an in silico ghr-miRNA-regulatory network was developed for CLCuKoV-Bur ORFs using Circos software version 0.6. These results represent the first predictions of ghr-miRNAs with the therapeutic potential for developing CLCuD resistance in upland cotton plants.

## 1. Introduction

Upland cotton, *Gossypium hirsutum* L. (allotetraploid), is an economically important agronomic crop grown for fiber and seed in many locations worldwide [1,2]. The first whole genome sequence of allotetraploid upland cotton (52 chromosomes (2n = 4x = 52)) and physical mapping were completed in 2015 [3]. Infection of the cotton crop in Pakistan and India by leaf curl disease (CLCuD) causes a reduced yield and reduced quality of the fiber and seed [4,5,6,7]. CLCuD is caused by whitefly-transmitted multiple species of begomoviruses, including cotton leaf curl Kokhran virus-Burewala (CLCuKoV-Bur) [6,8,9,10,11,12,13]. Begomoviruses are transmitted by the whitefly *Bemisia tabaci* (Gennadius) cryptic species complex (Hemiptera; *Aleyrodidae*) in a persistent-circulative manner [14,15,16,17]. CLCuKoV-Bur emerged as a virus and resistant breaking “strain” of CLCuD [7,10,13]. Begomoviruses have an ssDNA genome and can be monopartite or bipartite. They typically encode six proteins. The ORFs of CLCuKoV-Bur include a plus (+) virion-sense (VS) (V1 and V2) and negative (−) complementary-sense (CS) strand (C1, C3, and C4), respectively [7,10,18]. The large intergenic region (LIR) contains sequences involved in virus replication and regulation of transcription of the C1 and V1 ORFs of begomoviruses [19,20].

Despite multiple efforts to control the spread of CLCuKoV-Bur variants through improved whitefly management and introgression of resistance genes into *G. hirsutum,* the virus has continued to diversify and repeatedly overcome genetic resistance in cotton. RNA interference (RNAi) is a naturally occurring process in eukaryotes that mediates gene silencing in eukaryotes and, recently, has been harnessed as an invaluable research tool [21,22].

MicroRNAs (miRNAs) are a class of non-coding, single-stranded RNA, 18–25 nucleotide long, that are involved in the regulation of fundamental processes, including gene expression by silencing protein synthesis [23]. They are modulators of cell growth, differentiation, development, and antiviral defense responses. They have been used experimentally to modulate genes in cells through the activation or suppression of protein expression from mRNA to better understand the function and also as a therapeutic agent for interfering with specific host–virus interactions [24,25,26,27].

Artificial microRNAs (amiRNAs) are considered the most effective, preferred antiviral defense that triggers the silencing of a target gene [28,29]. The use of amiRNA for modulating gene expression in eukaryotes [30] has facilitated the customizable, highly reversible gene inactivation of selected targets [31]. Thus, amiRNA can be used to produce high-efficiency gene silencing capabilities that can complement viral genome-editing approaches for plants [32,33]. Relatively small, highly efficient amiRNAs cloned in plasmid vectors have been developed to trigger post-transcriptional gene silencing to combat virus infections [34,35,36,37,38]. And, in a short time, the miRBase has become a substantial resource for annotating miRNA molecules. Recently, 80 conserved mature *G. hirsutum* genome-encoded ghr-miRNAs have been identified and annotated using the most updated miRBase v22 [39], which has made available a number of mature upland cotton miRNAs involved in regulating biotic (i.e., insect and fungal pathogens) and abiotic (salt and drought) stressors [40,41,42,43,44,45].

The objective of this study was to develop an innovative, integrated approach to identify high-affinity target-binding sites of mature cotton ghr-miRNAs with the potential to silence CLCuKoV-Bur target genes. The goal was accomplished through the in silico screening of mature miRNAs using an integrated miRNA prediction process that implemented the RNA22, psRNATarget, RNAhybrid, and TAPIR algorithms for the identification of homologous amiRNAs. The sequences are derived from the unbiased mapping of ghr-miRNA–mRNA target site interactions and represent the first step toward constructing amiRNA-expressing silencing plasmid vectors and tools for the transformation of upland cotton. The hypotheses reported here for upland cotton are expected to lead to the development of CLCuKoV-Bur resistance in cotton plants engineered to express the cotton-encoded miRNA candidates to exploit an amiRNA approach.

## 2. Materials and Methods

### 2.1. Upland Cotton (G. hirsutum) MicroRNAs and CLCuKoV-Bur Genome Retrieval

Eighty experimentally validated mature cotton microRNA molecules (*Gossypium hirsutum*-microRNAs) (ghr-miRNA156-ghr-miR7514) have been archived in the miRBase database, and the number of cotton ghr-miRNAs in the database is expected to continue to increase. These available mature cotton ghr-miRNAs have been identified and characterized from the pre-miRNA stem–loop structured hairpin precursor sequences, ghr-MIR156-ghr-MIR7514. Subsequently, the mature ghr-miRNAs (Accession IDs: MIMAT0005806-MIMAT0029164) (Appendix A (Appendix A)) and precursor sequences (n = 78) (Accession IDs: MI0005638-MI0024206) (Appendix A) were obtained from the miRBase v22 (http://mirbase.org/) for screening in this study [39]. The complete genome sequence of a representative CLCuKoV-Bur (Accession no. AM421522) was downloaded from the NCBI GenBank database (http://ncbi.nlm.nih.gov) [46].

### 2.2. RNA22 Algorithm

The RNA22 algorithm is available as a web-based tool to detect evidence of direct interactions between pairs of miRNAs–mRNAs [47]. The publicly available, user-friendly RNA22 web-based tool relies on a non-seed-based approach for identifying statistically significant sequence motifs or “patterns” that represent miRNA-binding sites with corresponding hetero-duplexes, available at http://cm.jefferson.edu/rna22v1.0/ (accessed on 27 April 2022). This tool is ideal for interactively exploring and visualizing miRNA target-site prediction, in context [48]. The maximum folding energy (MFE) was set as −15.00 Kcal/mol.

### 2.3. psRNATarget Algorithm

The psRNATarget is a small plant RNA target analysis software that uses complementary scoring (schema V2) to identify predicted miRNA targets [49]. The psRNATarget is a standard target prediction program with an inherently high sensitivity for the detection of inhibition patterns. The psRNATarget is designed for plant genomes and is available on the web server, http://www.zhaolab.org/psRNATarget/analysis?function=2 [50] (accessed on 18 March 2019). The mature miRNA and mRNA profiles were uploaded and employed for putative target predictions using the default criteria available in Schema v2 (2017 release), as follows: seed region: 2–13 (NT); HSP size = 19; default expectation threshold = 7.0.

### 2.4. RNAhybrid Algorithm

The RNAhybrid is available at a publicly available online web server (https://bibiserv.cebitec.uni-bielefeld.de, accessed on 26 July 2021) developed to computationally predict miRNA–target binding sites based on intermolecular hybridization [51]. The RNAhybrid algorithm identifies the energetically most favorable hybridization sites of a small RNA in a large RNA and uses a seed binding site as the key biological element for miRNA-binding site detection in the coding and non-coding regions. It has been updated with features that show promise for predicting non-canonical target sites in miRNA molecules. The algorithm was used to estimate the minimum free energy (MFE) of the miRNA and mRNA target pair using the MFE setting of −20.00 Kcal/mol.

### 2.5. TAPIR Algorithm

Tracking Any Point with Per-frame Initiation, or TAPIR (http://bioinformatics.psb.ugent.be/webtools/tapir, accessed on 24 July 2023), is a state-of-the-art plant algorithm designed to identify targets for plant miRNAs. TAPIR is used to estimate seed-based miRNA–mRNA interactions and has two available modes, “fast” and “precise”, to identify target sites [52]. Here, TAPIR was used in the “precise” mode to select the most highly specific miRNA–mRNA duplexes and calculate the free energy ratio of the duplex. The analysis used the standard (default) parameters, with a maximum score of 9.0, and the MFE ratio was set at ≥0.2.

### 2.6. RNAfold Algorithm

The consensus miRNA precursor (cotton) sequences were obtained from miRBase, and the optimal RNA secondary structures were predicted using RNAfold, a folding algorithm based on thermodynamic principles [53]. The analysis was conducted using the default settings.

### 2.7. RNAcofold Algorithm

RNAcofold (http://rna.tbi.univie.ac.at/cgi-bin/RNAWebSuite/RNAcofold.cgi, accessed on 27 December 2024) is a recently developed web-based algorithm designed to estimate the free energy (Δ*G*) of the miRNA–mRNA duplex [54]. It was included as a complementary program that disallows pseudoknot configurations and specifically evaluates the concentration dependency of dimerization. In this study, cotton sequences were scanned with RNAcofold using default settings.

### 2.8. Discovering Cotton Genome-Encoded miRNAs–Target Interaction

The Circos program is used to visualize multi-miRNA binding sites and target interactions. In this study, the Circos algorithm was used to map the ghr-miRNA–CLCuKoV-Bur-mRNA target site interactions using the default settings [55].

### 2.9. Statistical Analysis

The R statistical software (version 3.1.1) was used to analyze and visualize multidimensional cotton miRNA data (Appendix A) and depict the results as graphical representations [56].

### 2.10. CLCuKoV-Bur Genome Annotation

The complete CLCuKoV-Bur genome sequence validation and annotation of coding regions were carried out using AcaClone software (v 1.1.152) to produce the pDRAW32 plasmid map (http://www.acaclone.com/) (accessed on 20 August 2022).

## 3. Results

### 3.1. High-Probability miRNA Binding Sites in CLCuKoV-Bur Genome

The computational predictive approaches available for identifying prospective miRNA targets have faced major bottlenecks, and target predictions have often suffered from high false-positive rates. To address the potential shortcomings of individual algorithms that offer different strengths and weaknesses, the following four widely used algorithms were deployed in this study for identifying the putative target sites of ghr-miRNAs: RNA22, psRNATarget, RNAhybrid, and TAPIR (Figure 1). The ghr-miRNA sequences and annotations have been aggregated in the miRBase (v22). Eighty mature ghr-miRNA sequences were curated following their extraction from miRBase (v22). By implementing an in silico workflow, the biological data and miRNA prediction algorithms were integrated to identify hybridization sites of the ghr-miRNAs with the predicted potential to silence coding and non-coding sequences homologous to CLCuKoV-Bur genome/mRNA sequences (Figure 2). The CLCuKoV-Bur genome is a single-stranded circular DNA molecule of 2759 nucleotides in length with five coding regions (Figure 3). The cotton miRNA-binding landscape for CLCuKoV-Bur-encoded mRNAs was mapped, and miRNA-binding sites were identified in the context of the viral genome-encoded mRNA and non-coding sequences. The small size and ease of accurate annotation of a begomovirus genome make it highly amenable to the unambiguous mapping of multi-ghr-miRNA genome binding sites.

The number and locations of the predicted binding sites varied, depending on the predictive tools. For example, the RNA22 algorithm predicted more than ten target sites. A comparison of complementary cotton host and CLCuKoV-Bur DNA miRNA targeting sites predicted the putative binding of 14 ghr-miRNAs, with the potential to target 18 CLCuKoV-Bur genomic sites.

The psRNATarget identified 22 ghr-miRNAs, with the potential to target 23 CLCuKoV-Bur binding sites that were scored as highly “cleavable targets”. The RNAhybrid detected 78 ghr-miRNAs corresponding to 78 target sites, whereas the TAPIR algorithm identified 16 ghr-miRNA–target pairs. Several miRBase ghr-miRNAs predicted by the computational pipeline were found to bind the CLCuKoV-Bur genome with particularly high affinity and included ghr-miR396 (a, b), ghr-miR7488, and ghr-miR7493, which were predicted by three of the four algorithms, psRNATarget, RNAhybrid, and TAPIR. And two ghr-miRNAs, ghr-miR169b and ghr-miR399d, were predicted to bind virus genome sequences based on the consensus by all four tools (Figure 4, Figure 5 and Figure 6, Table 1 and Appendix A).

### 3.2. Coat Protein (CP) of CLCuKoV-Bur Genome

The coat protein (CP) of the CLCuKoV-Bur genome is encoded by the V1 ORF, located between the coordinates 292 and 1062 (770 nt). The CP is required for encapsidation, cell-to-cell movement, and the whitefly vector transmission of begomoviruses [57,58,59]. The results of the RNA22 analysis identified ghr-miR169 (a, b) (start site 692) and ghr-miR7512 (918) as the most optimal target ghr-miRNAs (Figure 6A).

The eight miRNAs predicted to target the V1 ORF were ghr-miR3476-5p (start site 766), ghr-miR7485 (736), ghr-miR7492 (a, b, c) (902), ghr-miR7500 (675), and ghr-miR7510 (a, b) (703, 806), based on psRNATarget analysis (Figure 6B), while RNAhybrid predicted the miRNA binding sites (n = 8) ghr (miR394 (a, b) (852), miR482b (582), miR7486 (a, b) (850), miR7490 (563), miR7492 (a, b, c) (708), miR7510 (a, b) (695, 786), and miR7512 (918) (Figure 6C). By comparison, the TAPIR algorithm identified the V1 predicted targets as ghr-miR394 (a, b) (432) and ghr-miR2949 (b, c) (854) (Figure 6D and Appendix A).

### 3.3. Predicted Targets for the V2 ORF of CLCuKoV-Bur

The V2 ORF of CLCuKoV-Bur (132–488 nt) overlaps with the V1 ORF (CP) and encodes a pre-coat protein that is required for symptom development [60,61,62]. No host miRNAs were predicted to have the potential to target the VI and V2 overlapping regions based on the RNA22 and TAPIR algorithms (Figure 6A,D). However, the psRNATarget algorithm identified a target in the V1 and V2 ORF overlapping region, consisting of ghr-miR7497 located at nucleotide coordinate 460 (Figure 6B), whereas RNAhybrid predicted multiple targets in this region that consisted of ghr-miR2950 (320), ghr-miR3476-5p (422), ghr-miR7485 (473), ghr-miR7497 (363), and ghr-miR7498 (371) (Figure 6C and Appendix A)).

### 3.4. Predicted Targets for C1 ORF of CLCuKoV-Bur

The C1 ORF of the CLCuKoV-Bur genome (1505–2596) encodes a replication initiator protein (Rep) on the complementary strand. Rep is a multi-functional begomovirus protein that is essential for the replication of the circular ssDNA genome [63,64,65]. The RNA22 algorithm predicted five miRNAs in the rep coding region consisting of ghr-miR156c (1751), ghr-miR393 (1743), and ghr-miR399 (c, d, and e) (1747) (Figure 6A). By comparison, the psRNATarget identified four ghr-miRNAs with the potential to target C1 ORF, including ghr-miR399 (d, e) (1749), ghr-miR7493 (2095), and ghr-miR7505 (2059) (Figure 6B).

Predicted candidate ghr-miRNAs with the potential to target C1 ORF were identified as ghr-miR160 (2054), ghr-miR162a (1682), ghr-miR167 (a, b) (1954), ghr-miR399 (c, d, e) (1747), ghr-miR7484 (a, b) (1549), ghr-miR7487 (1575), ghr-miR7495 (a, b) (1596), ghr-miR7499 (1677), ghr-miR7502 (1656), ghr-miR7503 (1664), ghr-miR7505 (2046), ghr-miR7506 (1804), and ghr-miR7514 (1830) (Figure 6C). Also, the TAPIR algorithm predicted two “potentially efficient” host ghr-miRNAs: ghr-miR399d (1749) and ghr-miR2949a-3p (2079) (Figure 6D and Appendix A).

### 3.5. Predicted Targets for C3 ORF of CLCuKoV-Bur

The C3 ORF of the CLCuKoV-Bur genome (1059–1463 nt, coordinates) encodes the begomovirus replication enhancer protein (REn) [59,66,67]. The RNA22 algorithm predicted the following binding sites in this coding region, ghr (miR169b (1448), miR390 (a, b, c), (1410), and miR7491 (1422) (Figure 6A).

Analysis of the C3 ORF by the psRNATarget algorithm identified binding targets ghr-miR396 (a, b) (1249) and ghr-miR7493 (1163) (Figure 6B). The following nine hybridization sites within cotton miRNAs were predicted by RNAhybrid analysis, ghr-miR164 (1409), ghr-miR390 (a, b, c) (1410), ghr-miR 393 (1402), ghr-miR396 (a, b) (1252), ghr-miR 2949 (a-5p, b, c) (1241), ghr-miR7488 (1443), ghr-miR7489 (1451), ghr-miR7493 (1351), and ghr-miR7494 (1299) (Figure 6C). Finally, the TAPIR algorithm identified four miRNAs: ghr (miR172 (1236), miR396 (a, b) (1249), and miR7493 (1163) (Figure 6D and Appendix A).

### 3.6. Predicted Targets for the C4 ORF of CLCuKoV-Bur

The C4 ORF of the CLCuKoV-Bur genome (2137–2682, nt coordinates) encodes this smallest viral protein. It has been attributed to multiple functions, which are mostly poorly understood. The best-characterized function is as a ‘symptom determinant’ [68,69,70]. The ghr-miR390 (a, b, c) and ghr-miR7486 (a, b) located at nt coordinates 2443 and 2499, respectively, were identified in the C4-C1 overlapping region by RNA22 (Figure 6A).

The following four cotton-encoded miRNA binding sites were predicted with high confidence by psRNATarget, ghr-miR169b, ghr-miR7486 (a, b), and ghr-miR7513 located at the nt coordinates, 2190, 2499, and 2551, respectively (Figure 6B). By comparison, the RNAhybrid algorithm predicted the following cotton ghr-miRNAs, ghr-miR 156 (a, b, d), ghr-miR166b, ghr-miR169 (a, b), ghr-miR172, ghr-miR398, ghr-miR482a, ghr-miR 827 (a, b, c), ghr-miR3476-3p, ghr-miR7500, ghr-miR7501, ghr-miR7504b, ghr-miR7507, ghr-miR7509, and ghr-miR7513 located at the begomovirus genome coordinates 2203, 2584, 2190, 2187, 2480, 2580, 2484, 2481, 2478, 2440, 2205, 2152, 2172, and 2328, respectively (Figure 6C).

Finally, two target binding sites of cotton ghr-miRNAs predicted by the TAPIR algorithm were ghr-miR156 (a, b, d) (2202, nt coordinate) and ghr-miR169 (a, b) (2191, nt coordinate) (Figure 6D and Appendix A).

### 3.7. Large Intergenic Region of CLCuKoV-Bur

The large intergenic region, or LIR, of begomoviruses harbors non-coding sequences that are required for CLCuKoV-Bur replication and for the transcriptional regulation of the V1 and C1 ORFs [19,20]. The psRNATarget predicted the miRNAs, ghr-miR2948-5p (start site 72) and ghr-miR7491 (11) (Figure 6B).

By comparison, the RNAhybrid algorithm predicted the following six miRNAs, ghr-miR479 (150), ghr-miR2948-5p (66), ghr-miR7491 (11), ghr-miR7504a (131), ghr-miR7508 (84), and ghr-miR7511 (47). Also, RNAhybrid predicted the following three ghr-miRNAs in the LIR/C4 overlapping region: ghr-miR399 (a, b) (2649) and ghr-miR2949a-3p (2625) (Figure 6C). No predicted target binding sites were identified in the CLCuKoV-Bur genome by the RNA22 and TAPIR algorithms (Figure 6A,D and Appendix A).

### 3.8. Consensus miRNAs Predictions

Finally, a combination of algorithms, or a consensus approach, was evaluated for identifying conserved seed-binding sites among the cotton miRNAs. Among the 80 ghr-miRNAs in miRBase v22, 18 ghr-miRNAs were identified by two or more of the four algorithms evaluated (Table 2). Nine target binding sites were identified by consensus of at least two of the four algorithms: ghr-miR (156 (start site a, b, d), 169 (a, b), 390 (a, b, c), 396 (a, b), 399 (c, d, e), 7486 (a, b), 7488, 7493 and 7512) at nucleotide (nt) positions 2202, 2190, 1410, 1249, 1747, 2499, 2588, 1163, and 918, respectively (Figure 7 and Table 2).

Of the 18 consensus ghr-miRNAs, two upland cotton-derived miRNAs, ghr-miR169b and ghr-miR399e, were predicted to target the viral genome at coordinates 2190 and 1747, respectively, and were detected by the consensus of three of the four algorithms implemented. Of the eighteen consensus ghr-miRNAs, one cotton-derived miRNA, ghr-miR399d, was predicted to target the viral genome at coordinate 1747, located within the C1 coding region, and was detected by the consensus of all the four algorithms implemented: RNA22, psRNATarget, RNAhybrid, and TAPIR (Figure 8 and Table 2 and Table 3). The efficacies of the consensus ghr-miRNAs, ghr-miR169b, ghr-miR399d, and ghr-miR399e targeting CLCuKoV-Bur genes were validated by the suppression of RNAi-mediated cleavage or translational inhibition (Table 4).

### 3.9. Predicted miRNA–Target Interaction

A Circos plot was compiled to enable the exploration and follow-up validation of potentially complex miRNA–mRNA interactions involved in the upland cotton-CLCuKoV-Bur-pathosystem (Figure 9)

### 3.10. Estimation of the Free Energy (ΔG) of Consensus miRNA–mRNA Pairs

The binding free energies (ΔG) of the three consensus ghr-miRNA–miRBS (miRBS: miRNA binding site) duplexes, calculated by RNAcofold, ghr-miR169b, ghr-miR399d, and ghr-miR399e, were identified as the most effective seed matches for two interacting miRNA and mRNA molecules (Table 5).

#### 3.10.1. Secondary Structure Predictions

The MFE hairpin secondary structure parameters for the three consensus pre-miRNA sequences, ghr-MIR169b, ghr-MIR399d, and ghr-MIR399e, were determined by RNAfold and revealed the optimal secondary structure of the consensus precursors (Table 6).

## 4. Discussion

CLCuKoV-Bur is a highly transmissible and virulent begomovirus of upland cotton that has emerged in cotton–vegetable-producing regions of Pakistan beginning in the mid-1990s. Since then, CLCuD has occurred in cotton annually, and the composition of the virus populations that have predominated at any one time has been dynamic, with new species and strains emerging, most likely in response to attempts to release resistant germplasm from breeding programs in Pakistan to control the disease. The CLCuKoV-Bur species was responsible for the second major CLCuD pandemic in cotton that resulted in reduced yield and quality of cotton and an overall decline in productivity [6,7,8,10,13,71,72].

Recently, in silico approaches have been developed and evaluated to facilitate the prediction of multi-miRNA-binding sites to elucidate and modulate plant host–virus interactions [73,74,75,76]. Here, 78 precursors and 80 miRNA mature molecules, available at miRBase (v22), were downloaded and analyzed accordingly to identify upland cotton miRNAs with predicted binding sites in the CLCuKoV-Bur genome. Four widely accepted, variably robust, predictive tools that consider different criteria for predicting viral binding site interactions involve plant host miRNAs. These four algorithms, RNA22, psRNATarget, RNAhybrid, and TAPIR (Figure 1), were selected to take advantage of their computational complementarity and ability to reduce false-positive predictions. The upland cotton genome sequence queried was found to encode 18 predicted consensus ghr-miRNAs with the potential for binding CLCuKoV-Bur targets and silence replication and other key steps in the infection cycle (Table 2).

Individually, the RNA22 algorithm produced the most robust evidence of a direct interaction between miRNA and mRNA pairs, which is based on analysis of Watson–Crick and G:U pairs in the seed region [48]. The high-confidence consensual cotton miRNAs with the potential to bind to CLCuKoV-Bur sequences are summarized in Table 2. Also, in a previous study, the CLCuKoV-Bur genome was analyzed using the miRanda algorithm alone. The analysis yielded a large number of predicted ghr-miRNA target sites on the genome, making it difficult to select the several most optimal sites for in planta validation [77]. Other studies have reported attempts to identify potential miRNA-binding sites to target plant–virus–host interactions by in silico predictions in anticipation of curbing infections. A number of studies have explored amiRNA approaches for silencing other begomoviruses toward combat disease development in the plant host [28,29,78,79], and to attempt to interfere with the circulative, non-propagative transmission by the whitefly vector [31,80].

Overall, among the most optimal binding sites or targets of consensus of ghr-miRNAs were ghr-miR169b and ghr-miR399 (d, e) (Figure 8) located in the CLCuKoV-Bur C1 gene and C4-C1 overlapping region that encodes a symptom determinant. Interestingly, although robust predicted binding sites were identified in the C1 ORF and C4-C1 overlapping region as targets of consensus ghr-miRNAs that may feasibly inhibit CLCuKoV-Bur replication, no such consensus ghr-miRNAs were identified in the CLCuKoV-Bur C3 or V1 ORFs, even though the latter genes are likewise indispensable for virus infections (Figure 8). Extensive studies have already been conducted to determine the high-confidence miRNA binding sites for the targeted pathogen genome to alter host–virus interactions based on in silico criteria [81,82,83,84,85,86,87,88,89,90,91,92,93,94,95,96].

Together, the four algorithms, RNA22, psRNATarget, RNAhybrid, and TAPIR, predicted that a consensus high-confidence binding site of ghr-miR399d (start site 1747) (Accession no. MIMAT0014350) is capable of silencing the viral Rep protein (Figure 8; Table 2, Table 3 and Table 4). Both the single and consensus algorithms identified ghr-miR399d as the most highly promising molecule (Table 2, Table 3 and Table 4).

The most robust interactions between the CLCuKoV-Bur genome and host miRNAs that are predicted to inhibit translations are summarized in Figure 9. Silencing of the C1 ORF that encodes a Rep-associated protein of CLCuKoV-Bur is anticipated to reduce viral replication. Among the predicted ghr-miRNAs, the ghr-miR399d target site exhibited the strongest MFE value. The MFE of a consensus target pair (ghr-miR399d) was variably estimated by the different algorithms as −17.80 Kcal/mol (RNA22), −22.50 Kcal/mol (RNAhybrid), and −18.20 Kcal/mol (RNAcofold) (Table 3), making the pair potential “true targets” [75,97,98].

The complement between miRNAs and targets was predicted by the psRNATarget and RNAhybrid algorithms (Table 4). The RNAhybrid is a powerful algorithm that identifies seed-based interactions. The MFE calculated by RNAcofold was in the acceptance range for the formation of amiRNA–target duplexes. The cotton consensus ghr-miR399d identified by analyses is predicted to target the CLCuKoV-Bur C1 ORF sequence for silencing (Figure 8). Interestingly, a previous study has shown that the ghr-miR399 regulates salt stress in cotton [99] by targeting the non-coding RNA GhlncRNA973, which is a predicted target of ghr-miR399 [100].

To date, no experimental data have been published specifically focusing on cotton miRNAs for targeting the CLCuKoV-Bur DNA genome. Consequently, here, an integrative computational approach was evaluated with the hope of increasing the robustness of the analysis through the cross-validation of individual and multiple algorithms. The goal was to reduce the number and increase the accuracy of predicted potentially lucrative ghr-miRNA targets by combining the results of individual, union, and intersection analyses. The algorithms were evaluated individually and in different combinations to arrive at a consensus binding site identification. This computationally integrative approach was designed to authenticate the predicted results at individual, union, and intersection levels. The union-level prediction is based on combined results of in silico tools to detect ‘true targets’. The sensitivity of the prediction was also enhanced due to decreased specificity. The intersection approach was based on the combination of two or more in silico prediction tools (Figure 4 and Figure 8) [101,102,103].

While promising intermolecular RNA–RNA interactions between ghr-miRNAs and the CLCuKoV-Bur ORF genome have been established, the next step is to design optimal amiRNA expression vectors that impart precise sequence specificity and minimize deleterious off-target effects.

The small size of amiRNA molecules makes both vector construction and amiRNA expression tractable for in planta validation to determine if CLCuKoV-Bur replication can be reduced substantially or inhibited altogether. Once validated, the goal is to deploy multiple or “stacked” amiRNA targets to achieve durable, high-level tolerance against CLCuKoV-Bur and, ultimately, other species and strains associated with the dynamic CLCuD complex that is endemic to cotton cropping systems in Pakistan and India. Finally, to circumvent the development of resistance-breaking species and strains likely to occur with the continuous cultivation of virus-resistant plants [104,105,106,107], the amiRNAs must be selected to target multiple sites and achieve maximum binding affinity and high specificity to the CLCuD begomoviral mRNAs or non-coding targets, together with the assurance of the stable inheritance of the miRNAs in resultant germplasm [30,33,34,108,109].

In a prior study, the complete genome was sequenced and annotated for an isolate of CLCuKoV-Bur from naturally infected cotton in Pakistan [10], and the bidirectional promoter of CLCuKoV-Bur was characterized to elucidate transcription mechanisms [19,20]. In this study, 80 ghr-miRNAs were identified that have homology (based on in silico predictions) with the CLCuKoV-Bur genome, and multiple ghr-miRNA-binding sites were identified. Among them, predicted binding sites of consensus ghr-miRNAs were found to reside in multiple consensus regions of the CLCuKoV-Bur genome (Table 4). Through the host miRNA and CLCuKoV-Bur interactions with viral sequences, both the suppression and induction of viral gene expression and/or altered regulation of non-coding regions may occur when a virus infection activates the host plant defense systems. The Circos software was used to link mature miRNAs to their target genes by producing a network for ease of visualizing the interactions (Figure 9). This is the first report of the predicted relationships between specific host miRNAs, and their sequence targets have been characterized to create a holistic regulatory network map using a suite of computational tools to predict miRNAs encoded in the upland cotton genome and their putative targets in the CLCuKoV-Bur genome. Given the exciting progress toward the design of miRNA constructs of the CLCuKoV-Bur genome to undermine replication and other key steps in the infection cycle within the cotton host plant virus genome, targeting is poised to become the next generation of therapy that exploits knowledge of miRNA interactions with viral targets.

## 5. Conclusions

CLCuD is caused by a complex of whitefly-transmitted begomoviruses that cause leaf curl disease in cotton that has reached epidemic levels throughout the last four decades, with the “Burewala” CLCuKoV-Bur evolving as one of the most prevalent species among multiple species and strains associated with the ongoing CLCuD pandemics that have resulted in major losses to the cotton crop in Pakistan. In this study, the first predicted relationships between specific host miRNAs and their sequence targets have been characterized to create a holistic regulatory network map using a suite of computational tools to predict miRNAs encoded in the upland cotton genome and their putative targets in the CLCuKoV-Bur genome. In this study, 80 ghr-miRNAs were identified that have homology (based on in silico hybridization) to the CLCuKoV-Bur genome, and multiple ghr-miRNA-binding sites were identified. Among them, ghr-miR399d (Accession no. MIMAT0014350), located at coordinate 1747 in the CLCuKoV-Bur genome, exhibited more binding to their targets and concluded as the top effective ghr-miRNA with the predicted potential to target the C1 ORF. The C1 is a potential site for the antiviral construct to bind and inhibit the virus replication. In addition to ghr-miR-399d, the consensus cotton miRNAs ghr-miR169b and ghr-miR399e were also positively validated with a stable thermodynamic binding energy. Definitive experiments are also needed to determine the binding strength of the predicted ghr-miRNAs in transgenic cotton plants. The low regeneration capability of upland cotton callus is another constraint to the development of CLCuD-resistant cotton plants. The intersection of the results obtained from these integrated miRNA prediction tools allowed for identifying novel targets for designing amiRNA-based constructs. Future work will focus on the validation of these promising cotton ghr-miRNAs to develop CLCuKoV-Bur resistance in cotton plants. We can suppose that an amiRNA-based targeted approach has been expanded for long-term resistance against CLCuKoV-Bur infections.

## Figures and Tables

**Figure 1 viruses-17-00399-f001:**
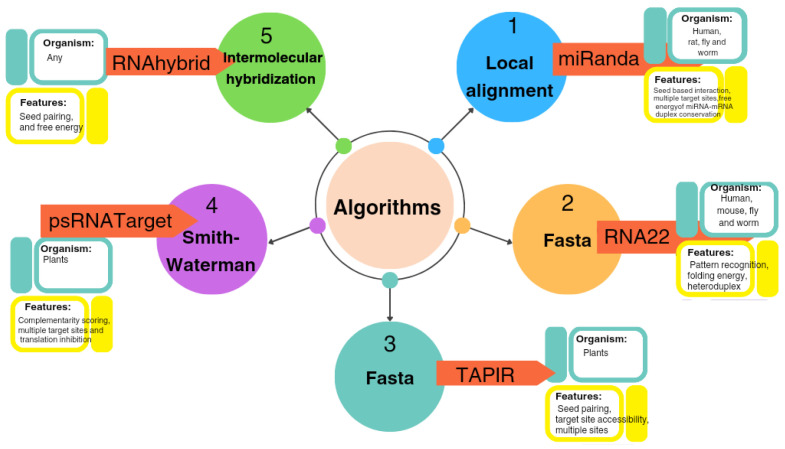
Features of common computational algorithms used for miRNA prediction. Four computational tools, RNA22, psRNATarget, RNAhybrid and TAPIR, were dedicated to showcase the predictive capacity of four widely used algorithms. The TAPIR and psRNATarget algorithms were used for plant miRNA predictions, whereas TAPIR and RNAhybrid are used to detect seed-based miRNA–target interaction. All four tools are multi-frame target detection algorithms and among the most widely used for analogous predictions.

**Figure 2 viruses-17-00399-f002:**
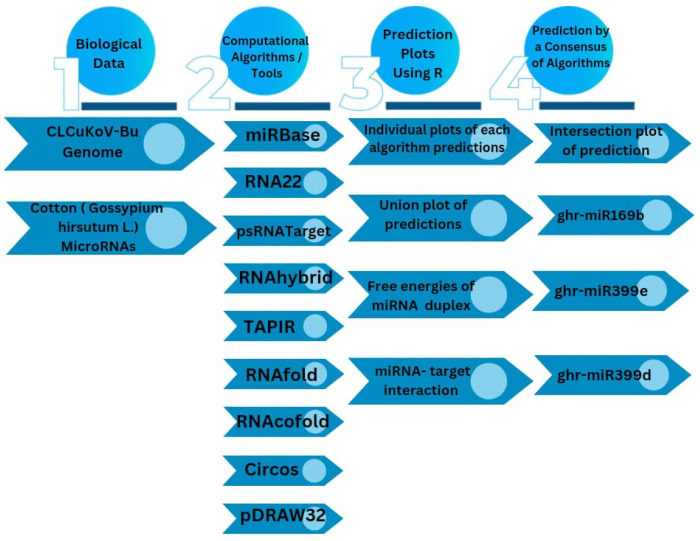
Schematic representation of selected miRNA target sites was designed in CLCuKoV-Bur genome that enables the integration of biological data and computational tools. The computational framework composed of six different types of in silico tools.

**Figure 3 viruses-17-00399-f003:**
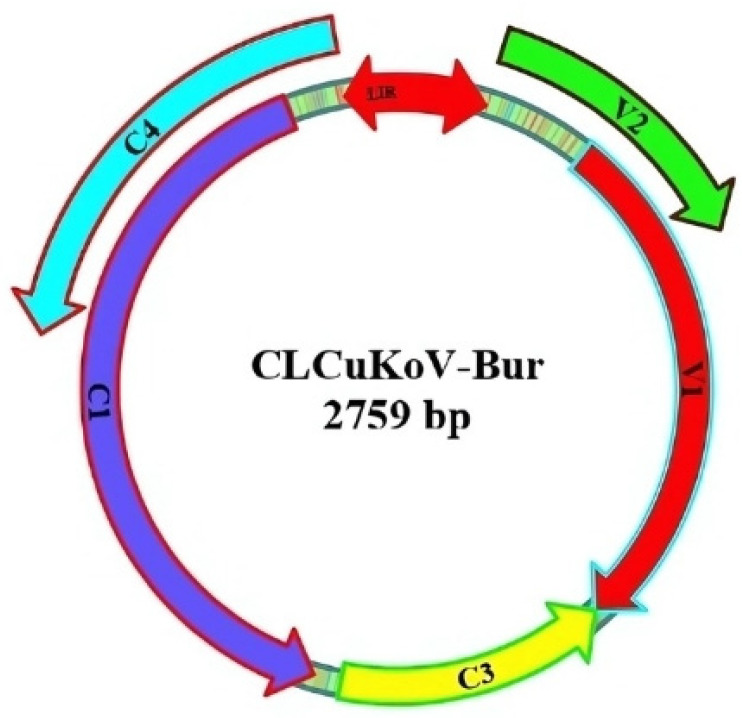
Diagram of the CLCuKoV-Bur genome sequence and coordinates for NCBI Accession no. AM421522. The predicted coding regions of the CLCuKoV-Bur genome indicated by the colored arrows.

**Figure 4 viruses-17-00399-f004:**
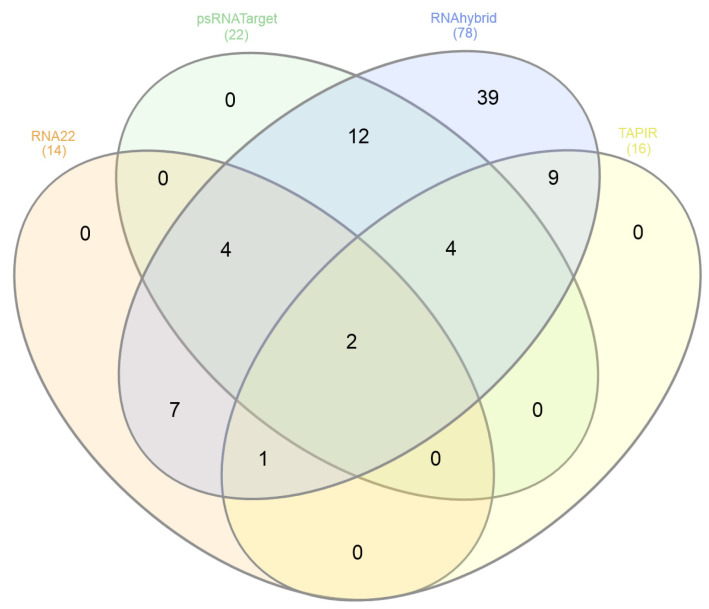
Venn diagram showing the number of ghr-miRNAs predicted by four algorithms RNA22, psRNATarget, RNAhybrid and TAPIR. The degree of overlap between in silico tool and multi-miRNA-binding site predictions.

**Figure 5 viruses-17-00399-f005:**
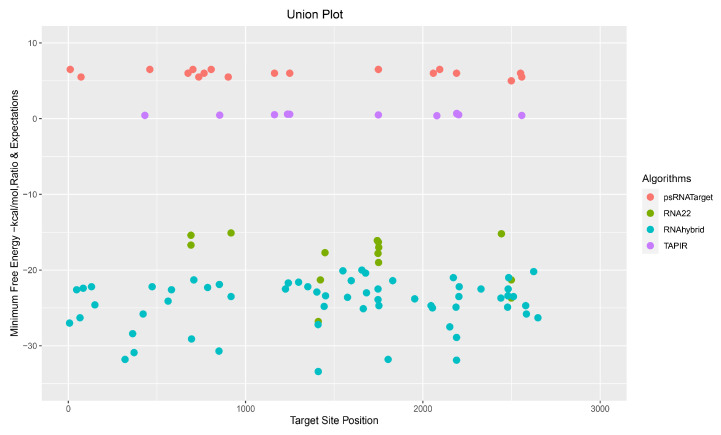
The union or consensus plot, showing the number of ghr-miRNA–target sites predicted by all four predictive tools. The y-axis shows the minimum free energy (MFE) based on −0–30 Kcal/mol threshold set for the RNA22 and RNAhybrid analyses. The MFE ratio for TAPIR was set over the range of 0–1, and the psRNATarget expectation score range was set at 0–10. The x-axis indicates the location in the virus genome sequence, spanning the 2759 nucleotides.

**Figure 6 viruses-17-00399-f006:**
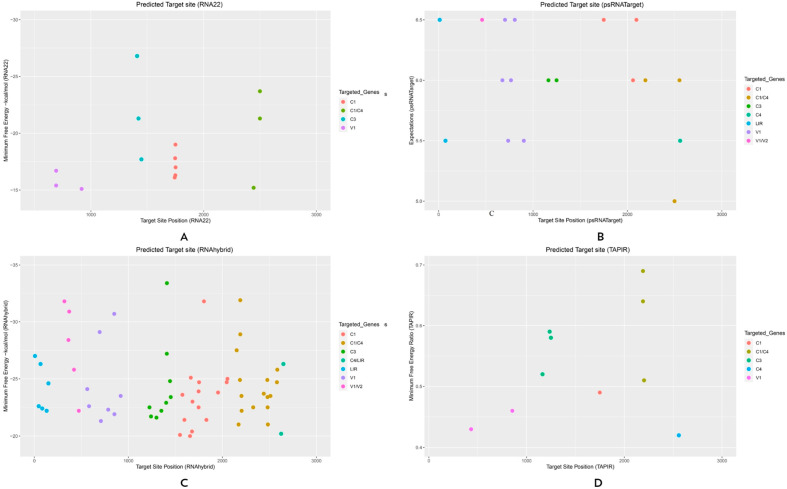
Individual cotton-encoded ghr-miRNA predicted to target CLCuKoV-Bur sequences in the ssDNA CLCuKoV-Bur. Four widely used miRNA predictive tools with high-dimension settings. The putative miRNA binding sites identified in the virus genome were predicted by (**A**) RNA22, (**B**) psRNATarget, (**C**) RNAhybrid, (**D**) TAPIR algorithms, respectively. The viral open reading frames that were identified as predicted targets are indicated by the colored dots.

**Figure 7 viruses-17-00399-f007:**
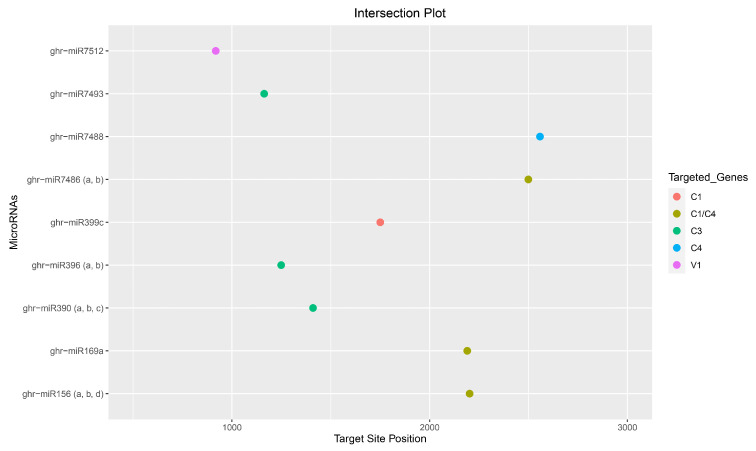
Intersection plot was created for ease of locating the consensus binding sites of cotton ghr-miRNAs. Nine high affinity predicted binding sites of 15 ghr-miRNAs in the CLCuKoV-Bur C1, C3, C4 and V1 coding region (s), identified by at least two of four algorithms implemented in this study.

**Figure 8 viruses-17-00399-f008:**
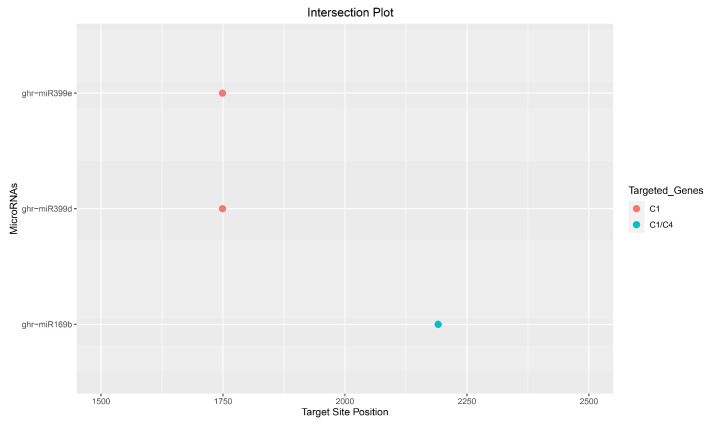
Intersection plot was created for ease of locating consensus binding sites of ghr-miRNAs. Two high affinity binding sites were identified to target different coding regions of CLCuKoV-Bur C1, and C4 by at least three of the four algorithms implemented. The x-axis indicates the genome position within the viral genome of 1–2759 nucleotides in length. The Y-axis indicates the consensus ghr-miRNAs.

**Figure 9 viruses-17-00399-f009:**
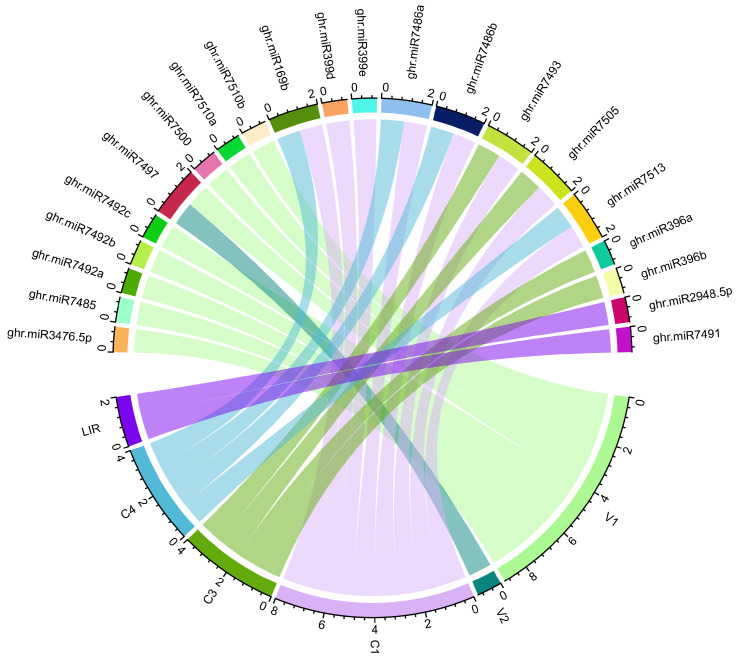
A Circos plot highlighting all predicted interactions between cotton ghr-miRNAs and CLCuKoV-Bur open reading frames (ORFs). This integrated interaction map shows miRNA–mRNA target interactions. The CLCuKoV-Bur ORFs are represented by colored lines.

**Table 1 viruses-17-00399-t001:** The number of cotton miRNAs predicted to target the CLCuKoV-Bur coding region.

Genes	Protein	RNA22 (Sites)	psRNATarget (miRNAs)	RNAhybrid (miRNAs)	TAPIR (miRNAs)
V1	Coat	3	8	12	4
V2	Pre-Coat	0	1	5	0
C1	Rep	5	4	18	2
C3	REn	5	3	14	4
C4	C4	5	5	19	6
LIR	-	0	2	10	0

**Table 2 viruses-17-00399-t002:** Summary of target binding sites of consensus ghr-miRNAs predicted by two or more of the four algorithms to target the CLCuKoV-Bur genome.

Cotton miRNA	Site/ORF RNA22	Site/ORF psRNATarget	Site/ORF RNAhybrid	Site/ORF TAPIR	MFE * RNA22	Expectation psRNATarget	MFE ** RNAhybrid	MFE Ratio TAPIR
ghr-miR156 (a, b, d)			2203 (C1/C4)	2202 (C1/C4)			−23.50	0.51
ghr-miR169a	692 (V1)		2190 (C1/C4)	2191 (C1/C4)	−16.70		−28.90	0.69
ghr-miR169b	692 (V1)	2190 (C1/C4)	2190 (C1/C4)	2191 (C1/C4)	−15.40	6.00	−31.90	0.64
ghr-miR390 (a, b, c)	1410 (C3)		1410 (C3)		−26.80		−33.40	
ghr-miR396 (a, b)		1249 (C3)	1225 (C3)	1249 (C3)		6.00	−22.50	0.58
ghr-miR399c	1750 (C1)		1752 (C1)		−19.00		−24.70	
ghr-miR3999d	1749 (C1)	1749 (C1)	1747 (C1)	1749 (C1)	−16.30	6.50	−22.50	0.49
ghr-miR399e	1747 (C1)	1749 (C1)	1747 (C1)		−17.80	6.50	−23.90	
ghr-miR7486 (a, b)	2499 (C1/C4)	2499 (C1/C4)	850 (V1)		−21.30	5.00	−30.70	
ghr-miR7488		2558 (C4)	1443 (C3)	2558 (C4)		5.50	−24.80	0.42
ghr-miR7493		1163 (C3)	1351 (C3)	1163 (C3)		6.00	−22.20	0.52
ghr-miR7512	918 (V1)		918 (V1)		−16.70		−23.50	

* MFE represents maximum free energy, calculated by RNA22. ** MFE denotes minimum free energy, estimated by RNAhybrid.

**Table 3 viruses-17-00399-t003:** The number of cotton miRNA-binding sites to target each gene of CLCuKoV-Bur.

MicroRNAs	RNA22	psRNATarget	RNAhybrid	TAPIR
	Folding Energy (*p*-Value)	Expectation	Minimum Free Energy	MFE Ratio
ghr-miR169b		6.00	−31.90	0.64
ghr-miR399d	−16.30 (0.319)	6.50	−22.50	0.49
ghr-miR399e	−17.80 (0.319)	6.50	−23.90	-

**Table 4 viruses-17-00399-t004:** The consensus cotton ghr-miRNAs were predicted for targeting (C1 and C4) genes.

miRNA ID	Accession ID	Mature Sequence (5′–3′)	Target ORF(s)	Genomic Target (nt)	Mode of Inhibition
ghr-miR169b	MIMAT0029157	CAGCCAAGGAUGAUUUGCCGG	C1/C4	2190–2212	Cleavage
ghr-miR399d	MIMAT0014350	UGCCAAAGGAGAUUUGCCCUG	C1	1747–1769	Cleavage
ghr-miR399e	MIMAT0025840	UGCCAAAGGAGAUUUGCCCCG	C1	1747–1767	Cleavage

**Table 5 viruses-17-00399-t005:** The binding interaction (ΔG) for the dimer cofold was estimated for each consensus pairs.

miRNA ID	miRNA–mRNA Sequence (5′–3′)	ΔG Duplex (Kcal/mol)	ΔG Binding (Kcal/mol)
ghr-miR169b	5′ CAGCCAAGGAUGAUUUGCCGG 3′ 5′ GCGGCGTAAGCGTCGTTGGCTGT 3′	−27.00	−19.15
ghr-miR399d	5′ UGCCAAAGGAGAUUUGCCCUG 3′ 5′ TGGACTGCCAGTCTCTTTGGGCC 3′	−18.20	−12.81
ghr-miR399e	5′ UGCCAAAGGAGAUUUGCCCCG 3′ 5′ TGGACTGCCAGTCTCTTTGGGCC 3′	−19.40	−14.88

**Table 6 viruses-17-00399-t006:** The stem–loop hairpin precursor for the three consensus ghr-miRNAs was characterized.

miRNA ID	Accession ID	Length Precursor	MFE */Kcal/mol	AMFE **	MFEI ***	(G + C)%
ghr-MIR169b	MI0024199	210 nt	−61.80	−29.42	−0.817	36.00
ghr-MIR399d	MI0013557	98 nt	−47.00	−47.95	−1.169	41.00
ghr-MIR399e	MI0022547	157 nt	−69.10	−44.01	−0.880	50.00

* MFE is minimum free energy. ** AMFE is the abbreviation of adjusted free energy. *** MFEI is defined as free energy index.

## Data Availability

The original contributions presented in this study are included in the article/Appendix A. Further inquiries can be directed to the corresponding authors.

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
