# Peer review of "An Integrative Computational Approach for Identifying Cotton Host Plant MicroRNAs with Potential to Abate CLCuKoV-Bur Infection"

_viruses, 2025, doi:10.3390/v17030399_

Round 1

Reviewer 1 Report

Comments and Suggestions for Authors

1. Although this paper is satisfactory overall, I am particularly concerned about the practical applicability of the bioinformatics work presented. The study primarily based on computational and in silico approaches, with no experimental validation of the predicted microRNA (miRNA) interactions or their effectiveness in suppressing CLCuKoV-Bur, which limits its practical relevance. I recommend that the authors include experimental work, such as Northern blot analysis, to confirm these interactions.

2. The authors could provide more discussion about any disparities or limits, such as high false-positive rates and sensitivity variations.

3. There are some typographical and grammatical errors, for example, in line 158, "Target prediction have suffered," which should be corrected to "Target predictions have suffered" or "Target prediction has suffered." Authors need to check others errors also. 

Author Response

RESPECTED REVIEWER 1 (GENERAL COMMENTS): 

Although this paper is satisfactory overall, I am particularly concerned about the practical applicability of the bioinformatics work presented. The study primarily based on computational and in silico approaches, with no experimental validation of the predicted microRNA (miRNA) interactions or their effectiveness in suppressing CLCuKoV-Bur, which limits its practical relevance. I recommend that the authors include experimental work, such as Northern blot analysis, to confirm these interactions.

Responses- Respected Sir, thank you very much for your kind comment. Discussion section has been greatly revised with experimental design and validation plan; Please see lines 455-469.  

General Comment-1: The authors could provide more discussion about any disparities or limits, such as high false-positive rates and sensitivity variations.

Response- Respected Sir, thank you very much for your kind comment. Discussion section has been greatly revised with more discussion on disparities or limits. Please see lines 168-173, 386-397 and 442-454.

General Comment-2: There are some typographical and grammatical errors, for example, in line 158, "Target prediction have suffered," which should be corrected to "Target predictions have suffered" or "Target prediction has suffered."

Response- Respected Sir, thank you very much for your kind comment. Correction has been made. Please see lines 169-170.

General Comment-3: Authors need to check others errors also. 

Response- Respected Sir, thank you very much for your kind comment. It has given us new direction to improve the manuscript. Correction have made throughout the manuscript.

Reviewer 2 Report

Comments and Suggestions for Authors

Remarks:

I guess that the presence of at least some experimental data (for example, agroinfiltration of leaves with a binary construct coding for ghr-miR399d microRNA followed by CLCuKoV-Bur infection) could improve this manuscript significantly

Figure 3: The quality of the drawing is not good enough (short transverse lines are out of focus)

Section “References”: Incorrect shortening of some journal titles, the use of capital letters instead of lowercase letters (see, for example, links:91, 96, lines 655-656, 665-666 as well as adjacent links 100 and 101). Please, check the entire list of references carefully.

Editorial notes:

Lines 57, 91, 115, 147: Not enough space(s)

Lines 71-73: “The amiRNA is highly reversible and flexible gene inactivation”. Please rephrase this and the following sentences

Line 82: Extra hyphen

Line 84: Should be plural for “algorithm(s)”

Line 116: Should be “putative” instead of “puative”

Line 164: Is italics for “algorithms” really necessary in this sentence?

Line 176: Should be italic for “in silico”, extra dot

Line 189: Should be “were compared” instead of “compared”

Line 202: Should be italic for “in silico”

Line 205: To my opinion it’s better to use “is” instead of “was” in this sentence. Please check the consistency of tenses throughout the text

Line 370: Should be italic for “in silico”

Author Response

I guess that the presence of at least some experimental data (for example, agroinfiltration of leaves with a binary construct coding for ghr-miR399d microRNA followed by CLCuKoV-Bur infection) could improve this manuscript significantly.

Responses- Respected Sir, thank you very much for your kind comment. Discussion section has been greatly revised with experimental design and validation plan; Please see lines 455-469.

General Comment-1: Figure 3: The quality of the drawing is not good enough (short transverse lines are out of focus).

Response- Respected Sir, thank you very much for your kind comment. Figure 3 has been reconstructed again: Please see in between lines 196-197.

General Comment-2: Section “References”: Incorrect shortening of some journal titles, the use of capital letters instead of lowercase letters (see, for example, links: 91, 96, lines 655-656, 665-666 as well as adjacent links 100 and 101). Please, check the entire list of references carefully.  

Response- Respected Sir, thank you very much for your kind comment. Correction has been made. All the reference has been revised with a uniform formatting.

EDOTORIAL NOTES: 

General Comment-1: Lines 57, 91, 115, 147: Not enough space(s).

Response- Respected Sir, thank you very much for your kind comment. Correction has been made.

General Comment-2: Lines 71-73: “The amiRNA is highly reversible and flexible gene inactivation”. Please rephrase this and the following sentences

Response- Respected Sir, thank you very much for your kind comment. Correction has been made. Pease see lines: 72-74

General Comment-3: Line 82: Extra hyphen. 

Response- Respected Sir, Correction has been made.

General Comment-4: Line 84: Should be plural for “algorithm(s)”

Response- Respected Sir, thank you very much for your kind comment. Correction has been made. Please see line 88.

General Comment-5: Line 116: Should be “putative” instead of “puative”. 

Response- Respected Sir, correction has been made. Please see line 123.

General Comment-6: Line 164: Is italics for “algorithms” really necessary in this sentence? 

Response- Respected Sir, correction has been made.

General Comment-7: Line 176: Should be italic for “in silico”, extra dot. 

Response- Respected Sir, correction has been made.

General Comment-8: Line 189: Should be “were compared” instead of “compared”

Response- Respected Sir, correction has been made

General Comment-9: Line 202: Should be italic for “in silico”. 

Response- Respected Sir, correction has been made throughout the manuscript.

General Comment-10: Line 205: To my opinion it’s better to use “is” instead of “was” in this sentence. Please check the consistency of tenses throughout the text. 

Response- Respected Sir, correction has been made. Please see line 216.

General Comment-11: Line 370: Should be italic for “in silico”. 

Response- Respected Sir, correction has been made throughout the manuscript.

Reviewer 3 Report

Comments and Suggestions for Authors

The manuscript by Ashraf et al. reports the presence of miRNAs in cotton that can confer resistance to Cotton leaf curl Kokhran virus-Burewala (CLCuKo-Bur). The authors found the potential therapeutic cotton genome-encoded miRNAs (ghr-miRNAs) using thorough in silico analyses of both cotton and CLCuKo-Bur genomes. The results presented here are informative and intriguing. However, explanations on the results were confusing in several places. For a better manuscript, I recommend that the authors pay attention to the following issues:

The statement between line #305 and line #307 described about the results shown in Figure 8 and Table 3. However, it seems that the authors did not explain the data very well. Table 3 highlighted three ghr-miRNAs that had potential binding sites on CLCuKo-Bur genome. Among the three, the authors emphasized the ghr-miR339d. Figure 8 also indicated the three miRNAs as potential therapeutic miRNAs. However, the description on the data stated that “only two miRNAs were predicted” (line #305) and that the ghr-miR399d was not included in the potential therapeutic miRNAs. Likewise, the results represented in Figure 9 and Table 4 were not explained very well (statement in lines #309-311).

I think that the ghr-miR399d should have been included in Figure 9. However, it was not there. Why is that?

In Table 2, the authors should explain what the MFE* and MFE** stand for.

The font sizes in Figure 6 are too small.

In Figure 7 and Figure 8, the y-axis looks longer than it needs to be. The computer-generated images do not look nice.

Some typos and grammatical errors were found.

Comments on the Quality of English Language

English is generally good, however, some typos and grammatical errors were found.

Author Response

RESPECTED REVIEWER 3 ( GENERAL COMMENTS): 

Comments and Suggestions for Authors

The manuscript by Ashraf et al. reports the presence of miRNAs in cotton that can confer resistance to Cotton leaf curl Kokhran virus-Burewala (CLCuKoV-Bur). The authors found the potential therapeutic cotton genome-encoded miRNAs (ghr-miRNAs) using thorough in silico analyses of both cotton and CLCuKoV-Bur genomes. The results presented here are informative and intriguing. However, explanations on the results were confusing in several places. For a better manuscript, I recommend that the authors pay attention to the following issues:  

General Comment-1: The statement between line #305 and line #307 described about the results shown in Figure 8 and Table 3. However, it seems that the authors did not explain the data very well.

Response- Respected Sir, thank you very much for your kind comment. Correction has been made. Please see lines 316-323 and 334-343.

General Comment-2: Table 3 highlighted three ghr-miRNAs that had potential binding sites on CLCuKoV-Bur genome. Among the three, the authors emphasized the ghr-miR339d. Figure 8 also indicated the three miRNAs as potential therapeutic miRNAs. However, the description on the data stated that “only two miRNAs were predicted” (line #305) and that the ghr-miR399d was not included in the potential therapeutic miRNAs.

Response- Respected Sir, thank you very much for your kind comment. Correction has been made. Please see lines 316-323 and 334-343. We have explained the results in two ways. First, I explained the results as 18 ghr-miRNAs are predicted on the basis of two algorithms (Table 2 and Figure 7). Then the results are explained on the basis of three algorithms. We have identified three miRNAs (Two on the basis of consensus binding sites predicted by three algorithms and one ghr-miRNA 399d was predicted all the four algorithms (Figure 8)

General Comment-3: Likewise, the results represented in Figure 9 and Table 4 were not explained very well (statement in lines #309-311).  

Response- Respected Sir, thank you very much for your kind comment. It has given us new direction to improve the manuscript. Correction has been made. Please lines 341-343.

General Comment-4: I think that the ghr-miR399d should have been included in Figure 9. However, it was not there. Why is that?

Response- Respected Sir, thank you very much for your kind comment. It has already been included in Figure 9.

General Comment-5: In Table 2, the authors should explain what the MFE* and MFE** stand for. 

Response- Respected Sir, thank you very much for your kind comment. It has given us new direction to improve the manuscript. Correction has been made as per reviewer suggestion. Please see table 2. Lines 332.

General Comment-6: The font sizes in Figure 6 are too small.

Response- Respected Sir, thank you very much for your kind comment. It has given us new direction to improve the manuscript. Correction has been made as per reviewer suggestion. Figure 6 has greatly improves

General Comment-7: In Figure 7 and Figure 8, the y-axis looks longer than it needs to be. The computer-generated images do not look nice. 

Response- Respected Sir, thank you very much for your kind comment. It has given us new direction to improve the manuscript. Correction has been made.

General Comment-8: Some typos and grammatical errors were found.

Response- Respected Sir, thank you very much for your kind comment. It has given us new direction to improve the manuscript. Correction has been made as per reviewer suggestion. The manuscript is greatly improved.

General Comment-9: English is generally good, however, some typos and grammatical errors were found. 

Response- Respected Sir, thank you very much for your kind comment. It has given us new direction to improve the manuscript. Correction has been made as per reviewer suggestion. The whole manuscript has been edited with highest standards of scientific editing.

Round 2

Reviewer 1 Report

Comments and Suggestions for Authors

The authors smartly avoided doing any experimental work, as I recommended in my first query. They haven't conducted any experiments, which severely reduces the validity of their findings. Experimental validation is essential to verify these interactions.

Author Response

The authors smartly avoided doing any experimental work, as I recommended in my first query. They haven't conducted any experiments, which severely reduces the validity of their findings. Experimental validation is essential to verify these interactions.

Responses- Thank you very much for your valuable suggestion. We agree on this concern but as this manuscript is only about screening the best possible microRNAs using an in-silico approach to create CLCuKoV-Bur resistance in cotton plants. The wet lab validation is not claimed by authors. The false-positive error rate was filtered using the in-silico tools described above to evaluate predicted miRNAs at three different levels. Definitive experiments are also needed to determine the binding strength of the predicted ghr-miRNAs in transgenic cotton plants. Our findings demonstrate the efficacy of RNAi-based transgenic cotton plants for CLCuD management. The application of RNAi to cotton varietal improvement to combat CLCuD infection offers a superior strategy for decreasing cotton yield loss. However, gene pyramiding for enhanced resistance to CLCuKoV-Bur for upland cotton is complicated because of the allotetraploid nature of the cotton genome. The low regeneration capability of upland cotton callus is another constraint to the development of CLCuD-resistant cotton plant. Future work will focus on the validation of this promising cotton ghr-miRNAs to develop CLCuKoV-Bur-resistance in cotton plants including evaluating the role of the predicted consensus ghr-miRNAs in CLCuKoV-Bur replication. It is further anticipated that the sequences predicted herein will be valuable for studying the mechanisms involved in host-virus interactions at the biological, genetic, and omic levels. Our manuscript creates a paradigm for highlight significant new advances in our understanding of plant–virus interactions that affect pathogenesis. We also hope to feature novel opportunities for controlling plant virus diseases that are now possible due to our understanding of plant virus pathogenesis. Correction has been made. Please see lines 509-512 and 514-515.

Reviewer 3 Report

Comments and Suggestions for Authors

The revised manuscript by Ashraf et al. has improved a lot. However, it seems that the manuscript still needs some corrections. The following issues could be considered:

  • Is the ‘upload’cotton in line #92 a typo for ‘upland’cotton?
  • In a phrase between lines #355 and #357, ‘Figure 9’ was mentioned two times, which was unnecessary.
  • In Table 6, the authors should explain what the MFE, AMFE, and MFEI stand for. In Table 2, the authors explain that the ‘MFE’ represents either ‘maximum free energy’ or ‘minimum free energy’. Which one does the ‘MFE’ in Table 6 represent?
  • It would be nice if the abbreviations ‘ghr’ (line #24) and ‘amiRNA’ (line #34) were explained when they first appeared.

Author Response

The revised manuscript by Ashraf et al. has improved a lot. However, it seems that the manuscript still needs some corrections. The following issues could be considered:  

General Comment-1: Is the ‘upload’cotton in line #92 a typo for ‘upland’cotton?

Response- Respected Sir, thank you very much for your kind comment. Correction has been made. Please see line 93.

General Comment-2: In a phrase between lines #355 and #357, ‘Figure 9’ was mentioned two times, which was unnecessary.  

Response- Respected Sir, thank you very much for your kind comment. Correction has been made. Please see line 356.

General Comment-3: In Table 6, the authors should explain what the MFE, AMFE, and MFEI stand for.  

Response- Respected Sir, thank you very much for your kind comment. Correction has been made. Please see line 377.

General Comment-4: In Table 2, the authors explain that the ‘MFE’ represents either ‘maximum free energy’ or ‘minimum free energy’.  

Response- Respected Sir, thank you very much for your kind comment. Correction has been made. Please see line 333.

General Comment-5: Which one does the ‘MFE’ in Table 6 represent?  

Response- Respected Sir, thank you very much for your kind comment. Correction has been made. Please see line 377.

General Comment-6: It would be nice if the abbreviations ‘ghr’ (line #24) and ‘amiRNA’ (line #34) were explained when they first appeared?  

Response- Respected Sir, thank you very much for your kind comment. Correction has been made. Please see lines 24 and 35.